# Establishing an early warning event management system at Africa CDC

Kyeng Mercy[1]*, Stephanie J. Salyer[1,2], Comfort Mankga[3], Calle Hedberg[3], Phumzile Zondo[3], Yenew Kebede[1]

1 Africa Centres for Disease Control and Prevention, Division of Surveillance and Disease Intelligence, Addis Ababa, Ethiopia, 2 United States Centers for Disease Control and Prevention, Division of Global Health Protection, Atlanta, Georgia; United States of America, 3 Health Information Systems Program South Africa, Pretoria, South Africa

* NjiT@africacdc.org

**Data Availability Statement:** All data analysed in this manuscript are included in Table 1 to Table 6.

**Funding:** The author(s) received no specific funding for this work.

## Abstract

Africa is home to hotspots of disease emergence and re-emergence. To adequately detect and respond to these health threats, early warning systems inclusive of event-based surveillance (EBS) are needed. However, data systems to manage these events are not readily available. In 2020, Africa Centres for Disease Control and Prevention developed an event management system (EMS) to meet this need. The district health information software (DHIS2), which is free and open-source software was identified as the platform for the EMS because it can support data capture and analysis and monitor and report events. The EMS was created through a collaborative and iterative prototyping process that included modifying key DHIS2 applications like Tracker Capture. Africa CDC started piloting the EMS with both signal and event data entry in June 2020. By December 2022, 416 events were captured and over 140 weekly reports, including 19 COVID-19 specific reports, were generated and distributed to inform continental awareness and response efforts. Most events detected directly impacted humans (69%), were considered moderate (50%) to high (29%) risk level and reflected both emerging and endemic infectious disease outbreaks. Highly pathogenic avian influenza, specifically H5N1, was the most frequently detected animal event and storms and flooding were most frequently detected environmental events. Both data completeness and timeliness improved over time. Country-level interest and utility resulted in four African countries adapting the EMS in 2022 and two more in 2023. This system demonstrates how integrating digital technology into health systems and utilising existing digital platforms like DHIS2 can improve early warning at the continental and country level by improving EBS workflow.

## Author summary

Disease surveillance data is critical for outbreak response, decision making and public health planning and program evaluation. The timeliness and completeness of this data both from indicator-based surveillance and event-based surveillance plays a critical role in the promptness of containing outbreaks and health emergencies at source. Digitizing

**Competing interests:** The authors have declared that no competing interests exist.

surveillance systems has shown to improve on the timeliness of surveillance data as well as the accuracy and completeness. While considerable investment has been made to digitize indicator-based surveillance data through the integrated disease surveillance and response strategy in some African countries, there still exist a significant gap in digitizing event-based surveillance systems. In addition, several countries are using the DHIS2 for their indicator-based surveillance, however this tool has not been exploited for event-based surveillance workflow. There are also challenges with interoperability as several tools deployed in countries are not interoperable with existing systems rendering data triangulation and deep analytics challenging. In this paper we show how the DHIS2 through an integration approach could be used to support event-based surveillance processes for prompt detection and reporting of health threats.

## Introduction

The African continent is home to hotspots for emerging and re-emerging diseases. In fact, all seven public health emergencies of international concern (PHEICs) declared as of 2023 have either emerged from or have severely impacted African countries. Zoonotic emerging infectious disease risk is especially elevated in forested tropical regions experiencing land-use changes and where there is high mammal species diversity [1]. With Africa losing an estimated 3.9 million hectares of forest a year between 2010 and 2020 [2] these trends in disease emergence are likely to continue. Africa is also home to 22 of the 25 most vulnerable national health systems in the world, as suggested by an analysis of infectious disease outbreak risk factors [3]. This indicates a need to strengthen early warning alert and response (EWAR) capacities in Africa at the continental, regional, country, and community level to timely and effectively detect and respond to health threats such as disease outbreaks.

Event-based surveillance (EBS) supports EWAR by detecting events of public health importance early and identifying the scope and magnitude of the events needed to inform response actions [4–5]. This was seen when epidemiologists took note of the first cases of haemorrhagic fever reported in Uganda in September 2022 [6] or when the first clusters of severe pneumonia were reported from Wuhan, China health facilities just prior to the eventual discovery of SARS-CoV-2 [7]. The Africa Centres for Disease Control and Prevention (Africa CDC) used the EBS modality of media scanning to monitor the spread of COVID-19 globally as well as the initial detection and progression of COVID-19 in Africa [8]. COVID-19 is not the last pandemic, and many global initiatives focus on early detection and prevention of the next pandemic.

Globally, disease surveillance has seen cycles of crisis and neglect for decades. The COVID-19 pandemic proved that low- and middle-income countries (LMICs) in Africa are at a particular disadvantage in a crisis compared to other continents, with intense competition and high/exorbitant prices for vaccines and personal protective equipment (PPE) as well as fewer resources to withstand economic and social shocks [9–10]. In addition to the EBS work Africa CDC is doing at the continental level, over 20 African countries implemented its EBS framework [11–12]. However, very few information systems exist to systematically record and manage EBS data—including the need to support critical EBS functions like monitoring the timeliness of detection, alert, and response.

In response to this need, Africa CDC in collaboration with the Health Information Systems Program South Africa (HISP-SA) developed an event management system (EMS) with initial funding from the African Union to support the implementation of EBS at the continental level

and also provide a solution to countries interested in improving their EBS related workflow. This article describes the system developed and its use in conducting EBS at Africa CDC, as well as its early adaptation in African countries.

## Methods

Prior to development, Africa CDC and HISP-SA identified the following system requirements, or objectives, for the EMS:

- **Is open source, web-based, mobile, adaptable, and an affordable platform.**

- **Supports standard EBS workflow,** inclusive of the five key EBS steps of detection, triage, verification, risk assessment, and alert described in the Africa CDC EBS Framework [13].

- **Allows for effective data capture and storage** of EBS-related data, including the ability to capture frequently used resource data (e.g., population, pathogen characteristics, socio-economic determinants of health, health sector capacity, and relevant research data).

- **Allows for data visualisation and analysis**, including the production of automated products tailored towards specific populations (e.g., social media messaging).

- **Produces and stores automated reports** (e.g., situation reports, briefs, etc.), inclusive of data visualisation like maps, tables, and charts.

- **Improves timeliness of EBS**, ensuring the inclusion of standardised monitoring and evaluation indicators like the Resolve to Save Lives 7-1-7 metrics [12,14].

- **Incorporates a One Health approach** by including human, animal, and environmental events that have a potential impact on public health.

- **Links to, integrates with, or can share data with other systems** to improve EWAR related data sharing and collaboration with African Union (AU) Member States, global and regional bodies, and partners. Systems include:

  ○ WHO's Epidemic Intelligence from Open Sources (EIOS) [15].

  ○ Country-level indicator-based surveillance (IBS) digital solutions like eIDSR for notifiable diseases that exceed routine reporting thresholds.

  ○ Data or data systems for emergency response teams, Emergency Operation Centres, national call centres, etc. to improve linkages to response efforts.

- **Is adaptable for use at the regional and country level**.

- **Assists with the use of EMS data for action** to inform policy, advocacy, response efforts and assess the impact of EBS.

The inclusion of these system requirements in the EMS development along with the collaborative and iterative prototyping approach used to develop it are described in this manuscript. We listed the DHIS2 applications used, modified, or created as part of the EMS development by the following application types: data entry, data visualisation and reports, and system configuration and maintenance. We further described the early implementation of the EMS by analysing aspects of the data recorded between June 2020 to December 2022. Data elements and indicators evaluated include staff using the system; number, type, and risk level of events detected by year; the most frequently reported events by year; and data completeness and timeliness indicators. For data completeness, we looked at the following variables: event start date, detection date (by Africa CDC and country), event reported date (by Africa CDC and

country), verification date, final risk assessment, lab result status, laboratory confirmation date and event intervention date (an intervention is defined as any response action taken by Africa CDC and country e.g., EOC activation, staff deployed, etc). Data, inclusive of both signals and events captured, were exported from the EMS into a comma-separated values (CSV) file and analysed in MS Excel. Signals are defined as the initial detection (by IBS or EBS) of a potential public health event, prior to verification. Events are "a manifestation of disease or an occurrence that creates a potential for disease" [13]; events can be infectious, zoonotic, food safety, chemical, radiological or nuclear in origin, and are transmitted by persons, vectors, animals, goods/food or through the environment. In the context of EBS and EWAR, an event refers to a signal that has been verified. For the country-level adaptations, we described the system attributes and indicators that were adapted, and the timeline associated with country level development and implementation.

No personally identifiable information was captured as part of this analysis. The United States Centers for Disease Control and Prevention (US CDC) deemed this project non-research and a routine public health practice.

## Results

### The collaborative Africa CDC EMS development

Between December 2019 and March 2020, a scoping review was conducted to develop the system requirements—identifying available systems, source documents, and existing Africa CDC EBS data being captured in Google and MS Excel spreadsheets. Africa CDC, US CDC Global Disease Detection Operations Center (GDDOC), WHO, the United Kingdom Health Security Agency (UKHSA, formerly Public Health England), Nigeria CDC (NCDC), AU Peace and Security, and European CDC (ECDC) staff were interviewed regarding their current EBS workflow and any existing event management systems used to support this work. Only GDDOC [16], ECDC [17], NCDC [18–19], and WHO [17] stated having an electronic EMS in place at the time of review; however, none of these were open source or readily available for adaptation by Africa CDC. DHIS2 was identified as the most suitable core platform for the EMS, for the following reasons:

1. Ability to cover system requirements identified during the scoping review, such as data capture, aggregation, analysis, and reporting.

2. Is open source with a large global community of developers and users; interoperable; and can be accessed/used in multiple ways (e.g., over the internet, by local intranet/locally installed system, via android mobile app) [20].

3. Is used as the main health management information system (HMIS) platform in over 76 countries globally, 48 in Africa [20].

4. Supports IBS (i.e., weekly aggregated and case-based) as part of HMIS implementation, providing an opportunity for system integration between EBS and IBS at the country level.

HISP-SA deployed a team of eight staff to collaboratively work with Africa CDC in developing the EMS. In May 2020, the initial prototype went live, and Africa CDC started capturing data into the system in June 2020. Weekly meetings were held between HISP-SA and Africa CDC to discuss development progress and share identified development needs. Communication included formal in-person meetings, video conferencing, emails, and other ad-hoc engagements. Small iterative updates were released for pilot testing at monthly intervals (Fig 1). HISP-SA held in-person trainings for Africa CDC EBS staff in November 2020 and 2021 on DHIS2 fundamentals and the use of the EMS. DHIS2 and EMS user manuals were

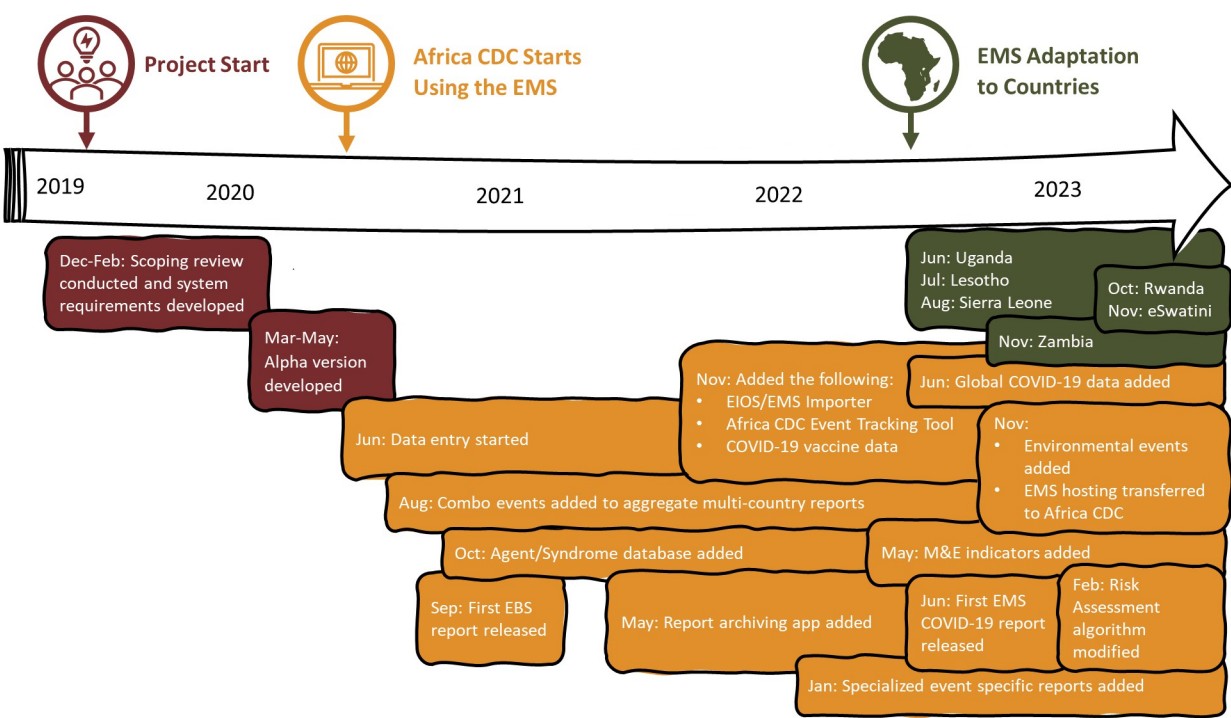

**Fig 1. Timeline of key Africa CDC Event Management System development milestones, December 2019 to December 2023.**

developed by HISP-SA—shared initially with Africa CDC in March 2021, and updated in February 2022. Hosting of the EMS was eventually transferred from HISP-SA to Africa CDC in November 2022.

As of December 2023, the EMS comprises 25 DHIS2 applications for data entry, data visualisation and reports, and system configuration and maintenance (Table 1). EMS data collection is mainly through the Tracker Capture application–one of the core DHIS2 applications customised to support the EBS workflow. All five EBS steps are linked to or supported by the Tracker Capture application (Fig 2):

- **Detection and triage:** allow for the recording of key EBS data related to detection; links to/ imports data from other applications/systems used to detect and triage signals (i.e., EIOS)

- **Verification:** allow for the recording of key EBS data associated with verification

- **Risk assessment:** has a semi-automated risk assessment component based on the Africa CDC algorithm [13]

**Table 1. DHIS2 applications used to develop the Africa CDC EMS as of December 2023.**

| DHIS2 application use categories | Level of application modification | | |
|---|---|---|---|
| | Not modified | Modified | Created |
| **Data Entry** | Bulk Load, Data Import/Export, Event Capture | Data Entry, Tracker Capture | EIOS/EMS Importer |
| **Data Visualization and Reports** | Dashboard, Data Visualizer, Event Reports, Event Visualizer, Line Listing, Maps, Standard Reports | | Africa CDC Event Tracking Tool, Africa CDC EBS Tracker Map |
| **Maintenance** | App Management, Browser Cache Cleaner, Data Administration, Data Store Management, Scheduler, System Settings, Usage Analytics, User Management | Maintenance | Africa CDC Dev Resources |

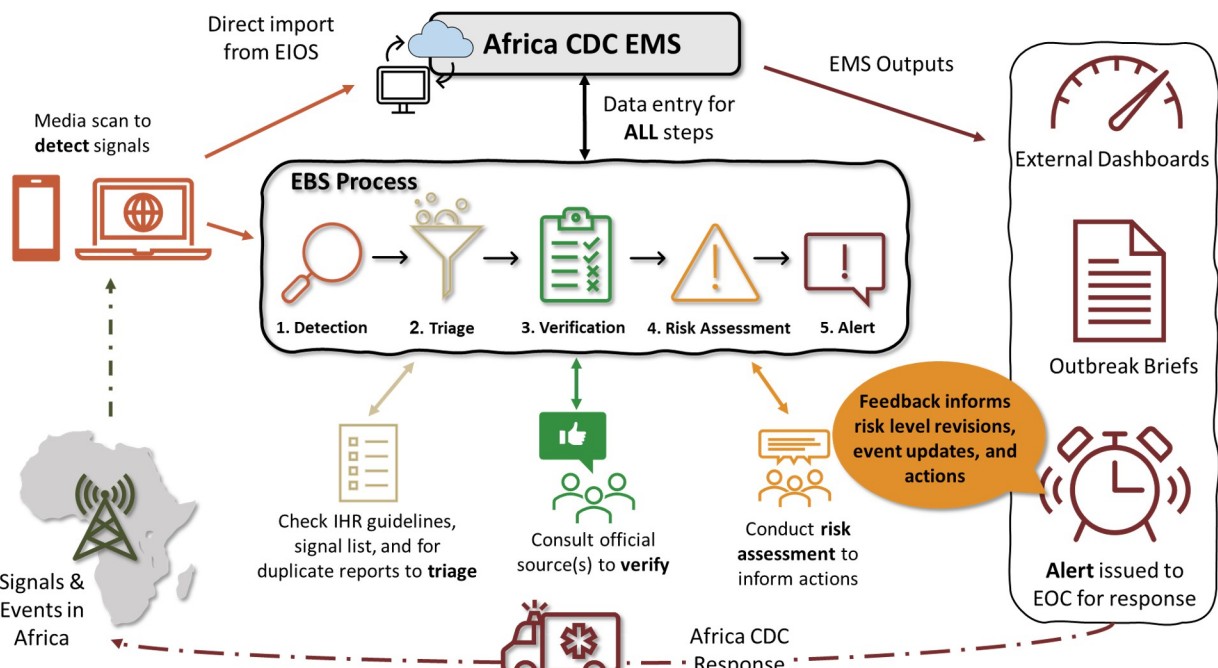

**Fig 2. Event-based surveillance workflow and process in relation to the Africa CDC Event Management System. Figure description: 1.** Signals and events occurring in Africa are detected by Africa CDC staff through media scanning efforts including the use of media aggregators like EIOS and social media like WhatsApp chat groups. Relevant EWAR information detected on EIOS is pinned to a specific board within EIOS for direct import into the Africa CDC event management system (EMS) via the **EIOS/EMS Importer application.** Signals are created within the EMS from imported articles. Information detected by other means is directly entered into the EMS via the **Tracker Capture application. 2.** Signals detected are triaged to ensure no duplicate reports are entered and that those entered meet IHR and Africa CDC EWAR criteria. Signals not fulfilling the triage criteria are closed out and discarded in the EMS using the **Tracker Capture application. 3.** Signals undergo a verification process by consulting official sources including any reports made by the local government. Once verified, a signal is changed to an event within the EMS and the date and source of the verification is documented in the **Tracker Capture application. 4.** All events undergo risk assessment and characterization by staff using the Africa CDC risk algorithm (see Africa CDC EBS Framework Chapter 1, Fig 4) (13). Responses to the risk algorithm questions and the date associated with the risk assessment are entered into the **Tracker Capture application**. The EMS automatically determines the risk level; however, staff have the option to override this determination by providing a justification statement. Risk assessment can be periodically updated as new information is received. **5.** Based on the risk level of an event, an automated alert is sent to key Africa CDC staff and EMS users via email to inform them of this event for any actions that need to take place. Weekly reports are also generated within the **Standard Report application**, exported, and then distributed (21) by Africa CDC staff to all stakeholders. Data entered can be analysed and visualised (e.g., dashboards, graphs, charts, maps, etc.) within the EMS through the **Data Visualizer, Dashboard, Maps, Standard Reports** and **Event Visualiser applications**. Additionally, EMS data can also be exported in CSV, CVS or XLS, JSON, HTML, and XML formats to feed external data visualisation applications. Routine monitoring, review, and updating of active signals and events is supported by the **Africa CDC Event Tracking Tool** with updates being made through the **Tracker Capture application**. Additionally, monitoring and evaluation of data accuracy and quality (not shown in figure) is supported with the use of the **Event Reports**, **Line Listing**, **Standard Reports** and **Dashboard applications**.

- **Alert:** automatically notifies key staff when high or very high-risk events are entered into the system, and links to other applications to support report generation

The Tracker Capture application records information about each event via 'profile' variables within the system. These profile variables are single variables associated with each event, such as the event start date, date of the initial response, or date of lab confirmation. There are also "non-profile" repeatable variables for each event, which can have multiple values associated with a specific date and time. For example, a risk assessment may change as a situation worsens or improves. Thus, recording these changes in risk level and the dates associated with the change are important to track. Similarly, other non-profile variables include cases, deaths,

lab tests conducted, and narrative information about response actions or geographic spread. That can be used to understand and report on the evolution of the event. Customization was done to allow for the automatic calculation of certain variables (e.g., cases and deaths) being captured to accommodate how some countries report data (e.g., cumulative cases vs. new numbers each week).

While any reporting interval can be used in the EMS, weekly reporting became and remains the standard for most events [21]. The COVID-19 pandemic became an exception as Africa CDC needed to enter data and report on this event in 55 AU Member States daily. Daily data entry through Tracker Capture was possible but became burdensome to staff. Thus, HISP-SA modified the core Data Entry application to accept a simplified version of aggregate COVID-19 data by country, saving significant data entry time.

Having a standard representation for the location of events is important. At first this "geographic anchor" (organisation unit) in the EMS was determined to be at the country level, with more granular location data being entered as free text. However, to capture data and report separately for multi-country outbreaks, a new "Africa Combo" organisation unit was created in August 2020.

To automatically calculate reporting statistics, worldwide population estimates per country, gender, and age group were imported from UN databases [22] and updated annually. A new "Agents/Syndromes" database within the Capture application was also created in October 2020 to store semi-permanent data related to the pathogen/agent characteristics. This program content was informed by a similar list created by the US CDC GDDOC. By June 2021, this database included information on more than 200 agents/syndromes. This information is useful because it can auto-populate technical details in the data entry fields for events, as well as inform EMS users about important event elements like expected pathology, disease courses, etc., on which to build their future EBS competence.

Timeliness indicators developed within the EMS [12] were initially informed by the Salzburg Global Seminar and related papers [23], and more recently by Resolve to Save Lives 7-1-7 indicators [14]. Other important custom functionality added included developing novel applications to support event monitoring, linkages to other systems like EIOS, and the integration of business intelligence (BI) platforms. For instance, a custom application, the Africa CDC Event Tracking Tool, was built to support staff with the routine monitoring, review, and updating of active signals and events in a rapid and systematic fashion. Data are displayed in a tabular fashion that is sortable by key variables displayed including event type, region or country affected, analyst monitoring, and date of last update. The event/signal number is hyperlinked to the Tracker Capture instance for any needed modifications or updates. Additionally, the EIOS/EMS Import application was developed to import information from news articles identified as signals or events during routine media scanning with EIOS. As a user finds articles that meet a signal definition, they can pin this article to a specific board within EIOS for direct importation into the EMS—simplifying the data entry workflow and reducing potential transposition errors. One example of BI integration includes the exportation of EMS data to create interactive, external dashboards [24] and other products to provide situational awareness to AU Member States, the general public, and partners.

## Early system outcomes

Prospective EMS data entry officially started on 8 June 2020. The total number of staff using the EMS fluctuated over time; in 2020, there were six staff entering events into the EMS, in 2021, there were 13, and in 2022 there were eight.

**Table 2. Number of events detected by type, human risk level, and year enrolled between 8 June 2020 and 31 December 2022 for all African countries.**

| | Number of country-level events detected | | | | | | | | | | | | |
|---|---|---|---|---|---|---|---|---|---|---|---|---|---|
| Type | Human* | | | | Animal* | | | | Environmental | | | | Total |
| Human Risk Level | Very high | High | Mod | Low | Very high | High | Mod | Low | Very high | High | Mod | Low | |
| 2020 | 0 | 11 | 22 | 15* | 0 | 1 | 0 | 2* | N/A | | | | 93 |
| 2021 | 1 | 28 | 58 | 9 | 2 | 4 | 19 | 2 | 0 | 0 | 1 | 0 | 126 |
| 2022 | 1 | 48 | 91* | 5 | 0 | 6 | 24* | 3 | 4 | 4 | 9 | 3 | 197 |
| Total | 2 | 87 | 171 | 29 | 2 | 11 | 43 | 7 | 4 | 4 | 10 | 3 | 416† |
| | 289 (69%) | | | | 63 (15%) | | | | 21 (5%) | | | | |

*Six events involved both animals and humans (1 low risk in 2020 and 5 moderate risk in 2022)

†Forty-nine events were enrolled in the system without a risk assessment. Thirty-nine of these were SARS-CoV-2 events from 2020. The remaining events for 2020 included avian influenza, cVDPV2, malaria, and measles; for 2021 Rift Valley fever and an unknown agent; for 2022 H5N1 (2 events), Rift Valley fever, and pneumonic plague. Risk levels defined in Africa CDC EBS Standard Operating Procedures.

As of December 2022, 416 events were detected and entered into the EMS: 93 events in 2020, 126 in 2021, and 197 in 2022 (Table 2). Most events directly impacted humans (69%) and were considered a moderate (50%) to high (29%) risk level. Most signals detected were verified as events; 95% of signals detected in 2020 were verified, 86% in 2021, and 90% in 2022. The first internal weekly EMS report was generated on 9 September 2020. Seventeen reports were generated in 2020, 52 in 2021, and 52 in 2022. The first COVID-19 EMS report was generated and distributed on 16 August 2022 [25] and 19 reports were produced in 2022.

The four most frequently reported human events in Africa by year reflect both emerging (e.g., COVID-19 and mpox) and endemic (e.g., vaccine-derived polio, measles) infectious disease outbreaks (Table 3). Cholera and "unknown agent" were the only events that were detected more than once from multiple countries. For animal events between 2020 and 2022,

**Table 3. Top four most frequently reported human country-level events detected by year.**

| Year enrolled in EMS | Event | Number of times detected | Number of MSs reporting events | Risk level* | Total confirmed cases reported | Total confirmed deaths reported |
|---|---|---|---|---|---|---|
| 2020† | COVID-19 | 39 | 39 | VH‡ | 4,006,809 | 93,064 |
| | Measles | 12 | 12 | M | 2,872 | 1,391 |
| | Cholera | 6 | 6 | L | 13,313 | 516 |
| | Unknown agent | 5 | 4 | L, M, H | 284,247 | 99 |
| 2021 | Vaccine-derived poliovirus | 17 | 17 | M, H | 148 | 0 |
| | Measles | 12 | 12 | M | 11,416 | 993 |
| | Cholera | 11 | 9 | L, M, H | 2,602 | 4,941 |
| | Yellow fever | 7 | 7 | M, H | 178 | 81 |
| 2022 | Measles | 25 | 25 | M, H | 36,184 | 3,165 |
| | Vaccine-derived poliovirus | 17 | 17 | M, H | 517 | 0 |
| | Cholera | 15 | 13 | M, H | 61,482 | 2,227 |
| | Mpox | 13 | 13 | M, H | 1,385 | 278 |

*L = low, M = moderate, H = high, VH = very high risk level

†Only prospective EMS entries for 2020 were analysed, thus events detected by Africa CDC between 1 January and 8 June 2020 are not reflected here.

‡The risk assessment was missing for these country-level events; thus, we used the "combo" event risk assessment to reflect the level that was assigned to this multi-country event.

**Table 4. Data completeness associated with core EMS data elements by year signal or event enrolled.**

| Data variable assessed | Number (%) events with missing data by year enrolled in EMS | | |
|---|---|---|---|
| | **2020 (N = 93)** | **2021 (N = 131)** | **2022 (N = 208)** |
| **Event start date** | 6 (6%) | 4 (3%) | 1 (0%) |
| **Detection date (Africa CDC)** | 53 (57%) | 17 (13%) | 0 (0%) |
| **Detection date (by country)** | 4 (4%) | 6 (5%) | 0 (0%) |
| **Event reported date*** | 3 (3%) | 4 (3%) | 0 (0%) |
| **Event reported date (by Africa CDC)*** | 64 (69%) | 26 (20%) | 0 (0%) |
| **Verification date** | 27 (29%) | 20 (15%) | 2 (1%) |
| **Final risk assessment** | 43 (46%) | 2 (2%) | 4 (2%) |
| **Lab result status** | 27 (29%) | 24 (18%) | 29 (14%) |
| **Laboratory confirmation date** | 43 (46%) | 76 (58%) | 68 (33%) |
| **Event intervention (by Africa CDC) date** | 79 (85%) | 108 (82%) | 175 (84%) |
| **Event intervention (by country) date** | 35 (38%) | 67 (51%) | 35 (17%) |

highly pathogenic avian influenza, specifically H5N1, was the most frequently detected event. For environmental events in 2022, storms and flooding were the most frequently detected events.

To monitor data completeness, several core variables were reviewed between June 2020 and December 2022 (Table 4). For almost all variables assessed, there was an improvement in data completeness when comparing data entry for 2020 with 2021 and, subsequently, 2021 with 2022. For EBS timeliness, we noted that the percentage of events meeting key timeliness indicators increased from 11% in 2020 to 44% in 2022 (Table 5).

## Country level adaptation

To date, EMS adaptation has been completed in six countries: eSwatini, Lesotho, Sierra Leone, Rwanda, Uganda, and Zambia (Fig 1). At the outset of adaptation, it was crucial to examine the similarities and differences in EBS data collection, interpretation, and reporting compared to Africa CDC. For example, a thorough understanding of the data platforms used for IBS and other surveillance data collection, as well as what tools are used for EBS (e.g., media scanning, hotline, etc.,) is required to be able to efficiently link existing data directly into an EMS. Integration with other systems meant working with other vendors/service providers and required

**Table 5. Timeliness associated with country-level events detected in Africa by year between 8 June 2020 to 17 March 2023.**

| Indicator assessed | Year of event detection | | |
|---|---|---|---|
| | **2020 (N = 93)** | **2021 (N = 126)** | **2022 (N = 197)** |
| **Timely event detection:** Proportion of events detected at Africa CDC within 7 days of event start | 16% (n = 15) | 29% (n = 37) | 41% (n = 80) |
| **Timely event verification:** Proportion of events which were verified by Africa CDC with an official source within 24 hours after detection | 22% (n = 20) | 44% (n = 56) | 71% (n = 139) |
| **Timely risk assessment for events:** Proportion of events for which the first risk assessment was conducted by Africa CDC within 24 hours of verification | 5% (n = 5) | 29% (n = 36) | 43% (n = 84) |
| **Timely event notification:** Proportion of events where Africa CDC issues an alert within 24 hours after verification | 10% (n = 9) | 36% (n = 47) | 73% (n = 152) |
| **Timely initial response:** Proportion of events for which a response was initiated by Africa CDC within 7 days of notification | 2% (n = 2) | 8% (n = 10) | 12% (n = 24) |
| **Timely initial response for high and very high risk events:** Proportion of high and very high level risk human events for which a response was initiated by Africa CDC within 7 days of notification | 8% (1/12) | 3% (1/35) | 22% (14/63) |

**Table 6. DHIS2 Applications and customizations associated with the country-level EMS adaptation.**

| DHIS-2 Applications and EMS Customizations | Adaptations Required | Features Adapted | Country |
|---|---|---|---|
| Africa CDC Dev Resources, App Management, Browser Cache Cleaner, Bulk Load, Data Administration, Data Entry, Data Import/Export, Data Store Manager, Data Visualizer, Dashboard, Event Capture, Event Reports, Event Visualizer, Line Listing, Maintenance, Maps, Scheduler, Standard Reports, System Settings, Usage Analytics, User Management, Africa CDC Event Tracking | No | No | eSwatini, Lesotho, Rwanda, Sierra Leone, Uganda, Zambia |
| Tracker Capture | Yes | Deactivated a feature for auto calculating new or cumulative cases on capturing | eSwatini, Lesotho, Rwanda, Sierra Leone, Uganda, Zambia |
| EIOS/EMS Importer | Yes | Integration depends on a key linked to a profile. Each country requires a specific key linked to their profile. | eSwatini, Rwanda, Sierra Leone, Uganda, Zambia |
| Custom HTML reports within the Standard Report application | Yes | Corporate theme adapted to countries and also adopt reports in use by countries | eSwatini, Lesotho, Rwanda, Sierra Leone, Uganda, Zambia |

expanding the Africa CDC EMS. From this thorough adaptation review, many country-specific customizations were made (Table 6), and some are highlighted below.

For all countries, the geographic organisation unit was changed from the country level to the subnational unit of district. The list of agents/syndromes was modified to be more specific and useful for each country, and a country-specific signal definition list was added as a new variable to the event profile in Tracker Capture. Report content and style were customised to meet each country's needs. The EIOS/EMS Importer was also modified to cover access permissions created on the EIOS system for respective countries by WHO.

The Africa CDC EMS also benefited from this adaptation process. Some countries wanted to monitor the timeliness of events in real time and record more specific outcomes related to event closure (e.g., after-action review outcomes, key event closure dates, outbreak, and response related summaries). Thus HISP-SA developed an indicator widget inclusive of 7-1-7 timeliness indicators and added a closing summary stage into Tracker Capture. The value added to the EMS from this adaptation was clear, and so it was added to the Africa CDC EMS system as well.

## Discussion

Prior to the EMS, Africa CDC used ad-hoc tools that were work intensive and required repetitious formatting and report production, reducing the time available for other critical EBS-related work. The EMS automated these tasks and enabled Africa CDC staff to collaboratively focus on detection, analysis, risk assessment, and response to events. The system allows for a multisectoral, One Health approach to EBS as it captures events affecting multiple populations and environments; this approach improves awareness of potential public health events like zoonotic outbreaks before they impact human health. EMS report generation has increased and diversified since inception showing the utility of this application and the potential for additional products [25].

Data entry completeness and timeliness metrics have improved over time. Completeness of key dates that rely on input only from the EMS users (i.e., the first 7 variables in Table 4)—as compared to those where the data is provided by sources outside the EMS system (e.g., lab)—are particularly striking from 2020 to 2022, with the proportion of missing key dates decreasing from an average of 31% to < 1%. Also, the percentage of events meeting the proposed timeliness indicators has increased from an average of 11% in 2020 to 44% in 2022. These

improvements from 2020 to 2022 are likely due to multiple factors like the adding of system prompts for data inaccuracies and increasing the skill level and experience of the EMS users over time through supportive supervision and multiple training sessions.

Ultimately Africa CDC has a goal to have at least 80% of events meeting the described timeliness indicators outlined in the EBS framework [12–13], and there is room to improve. Of particular concern is the low number of events, especially high and very high-risk events, where Africa CDC intervention dates are not recorded. The lack of data could indicate a true lack of action, or potentially highlight a need to link the EMS with the systems being used by the Africa CDC Emergency Preparedness and Response Division for monitoring response related activities. Improvement could be made by further incorporating automated alerts when critical fields are left blank or by creating mandatory data entry fields for critical dates. Supportive supervision and mentoring may also need to be strengthened to improve consistency across the team for routine EBS tasks—for instance, there was variability seen in the initial risk assessment level determined for the same disease (Table 3).

An automated M&E dashboard and report have been slated for the next phase of development. This will ensure real-time timeliness tracking, which could also help improve timeliness as an alert component could be set up to bring more attention to events not being addressed in a timely fashion. There is also a need to expand the current linkages of the EMS to other data sources. Specifically, there is an interest from both Africa CDC, and AU Member States in adapting the EMS, to link this system to other existing systems housing IBS data being collected as part of IDSR or other routine surveillance systems. Linking these data can help identify existing and predict future endemic disease outbreaks with the use of reporting thresholds overlaid on historic data. Other areas include linking to additional social media applications, laboratory and genomics data sets, and big data to expand the detection aspects of the EMS. To address the long-term maintenance and sustainability of the EMS, following the development, HISP-SA conducted a knowledge transfer to the Africa CDC system managers. At the country-level, countries have set up country teams with expertise in managing DHIS2 who now support country customization and maintenance.

Since the EMS was developed, other systems have been created or expanded their use case to incorporate event management. These include systems like EMS2 developed by WHO for use by WHO headquarters and regional office staff. This system is marked for the eventual expansion to the country-level in the coming years. SORMAS is also another open-source system initially developed to support IBS related data entry but has since been evaluated and adapted to support EBS data and event management [26–27]. Given that Africa CDC has set a goal to establish EBS, possibly using a variant of the EMS, in most AU Member States by 2030, it will be important to be aware of and in alignment with other EMSs established in Africa. Continued interoperability and standardisation of key EBS data elements will be critical for successful integration and regional collaboration around event detection and response in Africa.

## Conclusion

Africa CDC developed and implemented an EMS that would help streamline the existing EBS workflow and improve continental and AU Member State EWAR. While there is still much more work that needs to be completed to ultimately envision this goal, most initial objectives have been addressed with the advent of the Africa CDC EMS. The EMS provides an EBS data management solution not previously available that can be adapted to support other global regions and countries. It is in alignment with and supports Africa CDC's effort to strengthen data standards and data sharing across Africa's disease surveillance community, and the

system is expected to be steadily improved over the coming years. Areas considered for EMS expansion include strengthening linkages to IBS and other data sources to improve overall epidemic intelligence and early warning capacity. Additionally, the inclusion of more response related functions within the EMS could improve the utility and use for response related staff and thus the communication between and timeliness associated with surveillance and response efforts. With the apparent timeliness limitations posed by conducting EWAR activities at the continental and global level that are so dependent upon country-level data, building adaptable open-source systems like this will only improve EWAR overall as they are further established at a national level. Thus, advocacy and support for the continual establishment and improvement of systems like this are desperately needed.

## Acknowledgments

The authors would like to thank the US CDC GDDOC (Ray Arthur, Christine Manthey, James Fuller)), WHO (Johannes Christo Schnitzler (EIOS)), UKHSA (Ashley Sharp, James Elston (SITAware)), Nigeria Centers for Disease Control (Womi-Eteng Oboma Eteng (SITAware)), European Commission Joint Research Centre (Luigi Spagnolo (EIOS)) and staff who were part of the initial consultation for the development of the Africa CDC EMS. In addition, the authors appreciate the entire epidemic intelligence team of the Africa CDC for their support in piloting the system and providing feedback for improvements.

## Author Contributions

**Conceptualization:** Kyeng Mercy.

**Data curation:** Kyeng Mercy, Stephanie J. Salyer, Comfort Mankga, Phumzile Zondo, Yenew Kebede.

**Formal analysis:** Kyeng Mercy, Stephanie J. Salyer, Comfort Mankga, Phumzile Zondo.

**Methodology:** Kyeng Mercy, Stephanie J. Salyer, Comfort Mankga, Calle Hedberg, Yenew Kebede.

**Project administration:** Phumzile Zondo.

**Software:** Comfort Mankga, Calle Hedberg.

**Supervision:** Kyeng Mercy, Stephanie J. Salyer, Yenew Kebede.

**Validation:** Kyeng Mercy, Stephanie J. Salyer, Yenew Kebede.

**Writing – original draft:** Kyeng Mercy, Stephanie J. Salyer.

**Writing – review & editing:** Kyeng Mercy, Stephanie J. Salyer, Comfort Mankga, Calle Hedberg, Phumzile Zondo, Yenew Kebede.

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
