## [Decision Letter · Decision Letter 0]

7 Mar 2024

PDIG-D-23-00468

Establishing an early warning Event Management System at Africa CDC

PLOS Digital Health

Dear Dr. TETUH,

Thank you for submitting your manuscript to PLOS Digital Health. After careful consideration, we feel that it has merit but does not fully meet PLOS Digital Health's publication criteria as it currently stands. Therefore, we invite you to submit a revised version of the manuscript that addresses the points raised during the review process.

Please submit your revised manuscript within 60 days May 06 2024 11:59PM. If you will need more time than this to complete your revisions, please reply to this message or contact the journal office at digitalhealth@plos.org. Please include the following items when submitting your revised manuscript:

We look forward to receiving your revised manuscript.

Kind regards,

Gilles Guillot

Academic Editor

PLOS Digital Health

Journal Requirements:

Additional Editor Comments (if provided):

While the manuscript presents interesting material, the reviewers have identified many aspects on which it should be improved.

If you decide to submit a revised version, please address all the points raised.

Reviewers' comments:

Reviewer's Responses to Questions

**Comments to the Author**

1. Does this manuscript meet PLOS Digital Health’s publication criteria? Is the manuscript technically sound, and do the data support the conclusions? The manuscript must describe methodologically and ethically rigorous research with conclusions that are appropriately drawn based on the data presented.

Reviewer #1: Yes

Reviewer #2: Yes

2. Has the statistical analysis been performed appropriately and rigorously?

Reviewer #1: Yes

Reviewer #2: Yes

3. Have the authors made all data underlying the findings in their manuscript fully available (please refer to the Data Availability Statement at the start of the manuscript PDF file)?

Reviewer #1: Yes

Reviewer #2: Yes

4. Is the manuscript presented in an intelligible fashion and written in standard English?

Reviewer #1: Yes

Reviewer #2: No

5. Review Comments to the Author

Reviewer #1: This is an interesting article that documents a real world experience of conceptualizing an operationalizing an open source data platform for event-based surveillance. I have the following comments:

1) Article should be formatted according to the PLOS Digital Health submission guidelines. Please review them carefully and adhere to them.

Some examples - 

a) one of the authors has mentioned their educational qualifications in the title page - this is incorrect

b) references should not be cited in superscript in the paper

c) references should be in Vancouver style

d) author summary should be included

2) Abstract is too short - it does not summarize the paper well. The journal has a 300 word limit for the abstract, currently it is only 131 words, so you have scope to expand it further.

3) The paper, while well written, seems to be written in the style of a project report rather than a manuscript.

Some examples - 

a) lines 55 to 57 "We describe the system developed and its use in conducting EBS at Africa CDC, as well as its early adoption in African countries" can be updated to "This article describes the system developed and its use in conducting EBS at Africa CDC, as well as its early adoption in African countries".

b) Lines 42 to 43 - "We realize COVID-19 is not going to be the last pandemic" can be replaced with "It is well understood that COVID-19 will not be the last pandemic" and you can also cite a reference to support this - https://www.orfonline.org/research/the-covid19-pandemic-why-it-wont-be-the-last

4) Lines 27 to 29 - "This is not unexpected given that around 80% of global biodiversity, including pathogen diversity, is found in tropical regions". Is there a reference for this bold statement? If yes, please cite.

5) The difference between "signal" and "event" is not clearly delineated.

6) Can the authors develop a figure to depict the workflow of the system and how it helps to facilitate EBS? How is data for an event entered into the system - manually by the team of users, or automatically from EIOS and other sources, or a mix of both options? What happens after the event is created ("detected") in the system? How is it verified? How is its risk assessed? how is it notified? How is the response initiated? This would be very helpful to ensure clarity for the reader.

7) Line 131 - "Small consumable modules were released for pilot testing at monthly intervals". What does this mean? please clarify.

8) Lines 162 to 163 - "Switching from weekly to daily reporting for such major epidemics/pandemics is seamless." This seems like you are praising the system. Kindly do not use such language in an academic paper. Instead, you can rephrase it as - "The DHIS2 software enabled a seamless shift from weekly to daily reporting for COVID-19".

9) Lines 172 to 173 - "In order to automatically calculate reporting statistics, worldwide population estimates per

country, gender and age group were referenced from UN databases, and updated annually". Is this referring to the World Population Prospects which is annually published by the United Nations Population Division? If yes, please mention it and cite it as a reference (https://population.un.org/wpp/).

10) Lines 183 to 184 - "Other important custom functionality added included the addition of Africa CDC’s corporate

colours to DHIS2..." Is the addition of Africa CDC's corporate colors to DHIS2 an important custom functionality? I would disagree and suggest that this be deleted. The other custom functionality, though, is important and should be retained (integration of EMS with BI platforms).

11) Please do a thorough review of the paper to eliminate errors in grammar and spelling.

Examples -

a) Line 215 - it should be "Eswatini", not "Estwatini".

b) Line 237 - the last word should be "for", not "of".

12) Lines 280 to 282 - "Given that Africa CDC has set a goal to establish EBS, possibly using a variant of the EMS, in most AU Member States by 2030, it will be important to be aware of and supportive of other EMSs established in Africa". Suggest replacing "supportive of" with "in alignment with".

13) Lines 290 to 291 - "The EMS provides a solution not previously available and is something readily available to other global regions and countries" - the meaning of this sentence is unclear, please review.

14) Lines 299 to 302 - I do not think this section is needed "Biography of first author", please delete it.

15) Please include an acknowledgements section - the efforts of the EMS users and the HISP developers and the country teams where EMS piloting is occurring need to be acknowledged.

16) Table 1 - For the May 2022 row, the description is "See framework M&E". What does this mean? Is there an annexe for framework M&E?

17) In Table 3 and Table 6, the total number of events that occurred in 2020, 2021 and 2022 are 93, 126 and 197 respectively. But in Table 5, they are 93, 131 and 208. How can these totals be different? Please review and reconcile these differences.

18) What is the source of funding for the EMS? Please mention it in the Introduction section in lines 53 to 55 - "In response to this need, Africa CDC in collaboration with HISP-SA developed an event management system (EMS) with funding from .......... to support the implementation of EBS at the continental level but also provide a helpful solution to countries interested in improving EBS related workflow."

19) Is there a sustainability plan in place to ensure the hosting and routine maintenance of the EMS even after cessation of donor funding? Please include a sentence about that in the Discussion section in the paragraph on next steps.

Reviewer #2: The manuscript is generally well written, and I would advise the manuscript to be published. 

One thing that should be improved is that the manuscript focuses mainly on the anecdotal story of how the system was set up and spends little time on the actual results of the system (tables 4, 5, and 6). In fact, only two paragraphs are focussed on this key part of the manuscript.

Also, I do believe that the manuscript should focus slightly more on how the system has supported early detection. The manuscript doesn't have any metric to understand how early detection was improved by using this system. For example, where are these alerts reported by other systems from member states? If the data for this is not available, which I understand can be difficult to source, a discussion section could be added to explain this. 

A few other points:

Abstract: -information provided by the abstract does not support the last sentence of the abstract

-You mention countries adapting, do you mean adopting?

Introduction:

line 26. Mention the number of PHEIC in Africa

Line 38: please refrain from qualifiers when referring to 'astute' epidemiologists

Line 42: please refrain from personal opinions and focus on facts or rephrase it so that is not an opinion

Line 49: here you introduce the EBS at the ACDC. Please introduce it before

Line 53: spell out HISP-SA

Line 104-105. I'm unsure this information is relevant to the reader

Line 163: seamless doesn't seem to be a measurable metric

Line 183-184: If I understand correctly, here you mention the colour of the dashboard? If so, please remove as this is likely not relevant to EWARN implementation

Table 2. 'Created de novo' please rephrase to created

Table 5: the last two indicators are not clear on what an intervention by ACDC or a country is. Please define

Table 6: indicator 4. Notified to who? Indicator 5: what constitutes a response?

6. PLOS authors have the option to publish the peer review history of their article (what does this mean?). If published, this will include your full peer review and any attached files.

**Do you want your identity to be public for this peer review?** For information about this choice, including consent withdrawal, please see our Privacy Policy.

Reviewer #1: No

Reviewer #2: No

---

## [Editor Report · Decision Letter 1]

17 May 2024

PDIG-D-23-00468R1

Establishing an early warning Event Management System at Africa CDC

PLOS Digital Health

Dear Dr. MERCY,

Thank you for submitting your manuscript to PLOS Digital Health. After careful consideration, we feel that it has merit but does not fully meet PLOS Digital Health's publication criteria as it currently stands. Therefore, we invite you to submit a revised version of the manuscript that addresses the points raised during the review process.

Please submit your revised manuscript within 30 days Jun 16 2024 11:59PM. If you will need more time than this to complete your revisions, please reply to this message or contact the journal office at digitalhealth@plos.org. Please include the following items when submitting your revised manuscript:

We look forward to receiving your revised manuscript.

Kind regards,

Gilles Guillot

Academic Editor

PLOS Digital Health

Journal Requirements:

2. Please provide separate figure files in .tif or .eps format only and remove any figures embedded in your manuscript file. Please also ensure that all files are under our size limit of 10MB.

Additional Editor Comments (if provided):

In many cases (introduction) the citation style used by the authors is of the form

(ref number) Claim....

This is unsual.

The common usage is Claim.... (ref number).

As requested earlier by one of the reviewer: please check the journal citation style and submit a revision that complies with it.

Handling this type of formatting issue is the responsibility of the authors. 

It was not appropriate to rely on the reviewers to perform this task.

Abstract:

- Sentence "DHIS2 was identified as the platform for

the EMS as it can support data capture, monitoring, analysis, and reporting of events." is unclear if one is not familiar with DHIS2

- Spell out EBS in its first occurence (abstract)
---

## [Editor Report · Decision Letter 2]

8 Jun 2024

Establishing an early warning Event Management System at Africa CDC

PDIG-D-23-00468R2

Dear MERCY,

We are pleased to inform you that your manuscript 'Establishing an early warning Event Management System at Africa CDC' has been provisionally accepted for publication in PLOS Digital Health.

Best regards,

Gilles Guillot

Academic Editor

PLOS Digital Health